

# The impact of vestibular dysfunction on falls and postural instability in individuals with type 2 diabetes with and without diabetic polyneuropathy

Ina Ejsing Hunnerup Jørgensen[1], Louise Devantier[2], Hatice Tankisi[3], Henning Andersen[1] and Karolina Snopek Khan[1]

[1] Department of Neurology, Aarhus University Hospital, Aarhus, Denmark
[2] Department of Otorhinolaryngology, Department of Clinical Medicine, Aarhus University Hospital, Aarhus, Denmark
[3] Department of Clinical Neurophysiology, Department of Clinical Medicine, Aarhus University Hospital, Aarhus, Denmark

## ABSTRACT

**Aim**. This study aimed to determine the association between vestibular dysfunction, falls, and postural instability in individuals with type 2 diabetes (T2D) compared to healthy control individuals and to examine the impact of diabetic polyneuropathy (DPN).

**Methods**. This cross-sectional study included individuals with T2D with DPN ($n = 43$), without DPN ($n = 32$), and healthy controls ($n = 32$). Cervical and ocular vestibular evoked myogenic potentials (VEMP) were recorded, and latencies and amplitudes were determined. DPN was diagnosed based on nerve conduction studies and clinical scores. Postural instability was examined using a static posturographic balance system and calculated as an instability index (ST). Falls were recorded retrospectively during the past year. Group comparisons were conducted by using univariate and bivariate statistics.

**Results**. Individuals with T2D experienced more falls than healthy controls (T2D with DPN $n = 12[38\%]$, T2D without DPN $n = 15[35\%]$, controls $n = 5[16\%]$, $p = 0.04$). Individuals with T2D had decreased postural stability, T2D with DPN, ST (median of 52[iqi = 33; 77]), T2D without DPN, ST (median of 31[iqi = 24; 39]), controls ST (median of 26[iqi = 19; 33], $p = 0.01$), when comparing all three groups. Individuals with T2D had a greater number of no-responses in oVEMP compared to controls (T2D with DPN, $n = 15[46.9\%]$ T2D without DPN $n = 25[58.1\%]$, controls $n = 9[28.1\%]$, $p = 0.04$). No difference was found in cVEMP and oVEMP amplitudes in any of the groups. Irrespectively of DPN, fallers with T2D had decreased oVEMP and cVEMP latencies on the right ears, when comparing to non-fallers, respectively, n10 (fallers [median of 16, iqi=15;19 ms.] *vs.* non-fallers [median of 25 iqi=16;35 ms]); p13 (fallers [median of 16, iqi=15;17 ms.] *vs.* non-fallers [median of 15, iqi=8;16 ms.], $p < 0.05$).

**Conclusion**. Falls and postural instability were more frequent in individuals with T2D compared to healthy controls. Fallers with T2D had vestibular end-organ impairments based on the oVEMP and cVEMP latencies on the right but not the left ears, irrespective of DPN. Individuals with T2D had more frequent no-response of the oVEMP, indicating impaired vestibular nerve function.

Corresponding author
Karolina Snopek Khan,
kasnop@rm.dk

## INTRODUCTION

Individuals with diabetes have an increased risk of falls and postural instability (*Yang et al., 2016*; *Khan et al., 2021*), which may result in impaired mobility, fall-related injuries, and increased mortality (*Tilling, Darawil & Britton, 2006*). To prevent falling and maintaining postural stability, proper functioning of the sensory, motor, visual, and vestibular system is required (*Hewston & Deshpande, 2015*; *Grace Gaerlan et al., 2012*). Individuals with type 2 diabetes have a higher incidence of vestibular dysfunction, especially individuals with diabetic polyneuropathy (*Li et al., 2019*; *Agrawal et al., 2009*; *Agrawal et al., 2010*; *Ward et al., 2015*). Therefore, the influence of vestibular system dysfunction and diabetic polyneuropathy on falls is of great importance.

Vestibular dysfunction may present as a subclinical vestibular neuropathy (*Konukseven et al., 2015*). In diabetes, the function of the vestibular system has been studied using cervical and ocular vestibular evoked myogenic potential (VEMP), including in individuals both with and without symptoms of vestibular dysfunction. VEMPS are short-latency reflexes recorded from ocular (oVEMPs) or cervical (cVEMPs) muscles. cVEMP reflects ipsilateral sacculus and inferior vestibular nerve function, whereas oVEMP reflects contralateral utriculus and superior vestibular nerve function (*Rosengren & Kingma, 2013*).

Studies have previously examined vestibular function in individuals with diabetes using VEMP, but with conflicting results. *Bektas et al. (2008)* found no difference in cVEMP responses between individuals with diabetes, with and without diabetic polyneuropathy (DPN), and healthy controls. Other studies found decreased cVEMP and oVEMP amplitudes in individuals with diabetes compared to healthy controls (*Ward et al., 2015*; *Kalkan et al., 2018*), whereas others found prolonged cVEMP (*Konukseven et al., 2015*; *Kamali et al., 2013*) and oVEMP latencies (*Konukseven et al., 2015*). Although a few studies have been conducted on vestibular function in individuals with diabetes using VEMP, no previous studies have investigated the impact of DPN on vestibular dysfunction and the possible association of falls and postural instability.

Therefore, the aims of our study were to assess the association between vestibular dysfunction, falls, and postural instability in individuals with type 2 diabetes compared to healthy control individuals and to examine the impact of diabetic polyneuropathy.

## METHODS

This cross-sectional study included data on secondary outcomes from a subpopulation of individuals evaluated and described in an earlier published study conducted at Aarhus University Hospital in Denmark between August 2017 and November 2018 (*Khan et al., 2021*). The study protocol was registered with the Danish Data Protection Agency (approval no.:1-16-02-563-16) and approved by the Central Denmark Region Committees on Health Research Ethics (approval no.: 1-10-72-282-16). Written informed consent was acquired

from all participants, and all work was done in accordance with the Declaration of Helsinki (1964).

Individuals with type 2 diabetes were included if they were between 18-80 years and had a diagnosis of type 2 diabetes based on the 1999 WHO criteria (*American Diabetes Association, 2011*). This is described in more detail elsewhere (*Khan et al., 2021*).

Participants were excluded if they had a history of transplantation or had complications relating to diabetes that could impact the examination of postural stability, falling, or DPN. The exclusion criteria were: amputation or severe deformity of the lower extremities, musculoskeletal disease, symptomatic osteoarthritis, peripheral vascular disease (including abnormal pedal pulses, cool skin, and abnormal skin color), stroke, ischemic heart disease, other causes of polyneuropathy, blindness, other concomitant neurological or endocrine diseases). The control group was composed of healthy volunteers who were recruited through local advertising. Healthy controls had normal glucose tolerance, blood pressure, and lipid profiles.

## DPN assessment

Individuals with diabetes were assigned to the DPN group if meeting the Toronto diagnostic criteria for confirmed DPN (*Tesfaye et al., 2010*), defined as an abnormality in nerve conduction studies (NCS, a symptom, and/or a sign of DPN based on the validated Toronto Clinical Neuropathy Score (*Bril & Perkins, 2002*). Motor NCS in peroneal and tibial nerves and sensory NCS in sural nerves, including the distal segment, were performed using standard surface electrodes techniques (*Tankisi et al., 2019*). The results were compared with laboratory controls. At least two abnormal nerves, one of which was the sural nerve, were required for abnormality in NCS (*Tankisi et al., 2019*).

## Clinical and biochemical assessment

A physician screened all participants, including evaluation of the previous medical history. Information concerning body height, weight, and waist circumference was collected, and body mass index (BMI) was calculated. Information on disease duration, use of insulin, and oral anti-diabetes agents was obtained. Furthermore, blood samples were collected and analyzed for HbA$_{1c}$. Data were collected as previously described in *Khan et al. (2021)*.

A physician assessed visual acuity using Snellen's test. Sway was measured at eight sessions of 32 s (eyes open''-/closed, on foam pads and on hard surface, head turned right and left, head up and head down) using a validated static posturographic balance system (Tetrax, IA, Israel) (*Khan et al., 2021*; *Christensen et al., 2018*). Participants were instructed to stand barefoot on a platform consisting of four independent force plates with their arms along their sides. The extent of sway over the four force plates was described by a stability index (ST), with a high ST value reflecting poor postural stability. Calculation of the ST value:

"ST $= t\{ \sum n\ 1[(an - na - 1)2 + (bn - bn - 1)2 + (cn - cn - 1)2 + (dn - dn - 1)2]\}1/2/ W.$"(*Gorski et al., 2019*). t $=$time (32 s), n $=$number of signals recorded, the four plates (a/b/c/d), W $=$total body weight.

A physician collected information on fall history and a fall was defined as "an event that results in a person coming to a rest unintentionally on the ground or another level" (*WHO,*

*2008*). All participants reported the frequency of falls over the past year. The physician ensured that all participants concurred on the definition of a fall excluding the following causes: vasovagal and cardiogenic syncopal episodes, hypoglycemia, and mechanical or external forces.

## VEMP

Eclipse EP25 Evoked Potential System (Interacoustics A/S, Middlefart, Denmark) was used for all examinations. Data on VEMPs were collected as previously described in *Brix, Ovesen & Devantier (2019)*. The electrodes were placed according to standarized protocols (*Interacoustics, 2022a*; *Interacoustics, 2022b*) by a trained physician (KSK) that performed all the examinations. Before attaching the electrodes, the skin was carefully cleansed, securing a skin impedance below 10k $\Omega$. 200 stimuli per trial were averaged. A minimum of two trials and one control trial were run. The analysis interval was 100 ms including a 20 ms prestimulation interval. VEMP was considered absent (no response) when a biphasic waveform was missing.

### Cervical VEMP (cVEMP)

Participants were seated upright with the head rotated opposite to the side of the stimulation. The active electrode (Neuroline 720 Single Patient Surface Electrodes, 8500060; Ambu, Ballerup, Denmark) was placed on the upper third part of the sternocleidomastoid muscle. The reference electrode was placed on the jugular notch, and the ground electrode on the forehead.

In order to minimize the impact of muscle contraction on the cVEMP amplitude, symmetry and reproducibility, electromyography information was displayed on a computer monitor for the patient to maintain a tonic contraction. The software was set to only stimulate and thus record if the tonic contraction sternocleidomastoid muscle was between 50–150 $\mu$V (*Interacoustics, 2022a*; *Kumar, Bhat & Varghese, 2018*). EMG scaling (amplitude correction) was performed using EMG magnitude estimates obtained from the mean root mean square (RMS) of the EMG occurring before stimulus onset. In-earphone plugs (3M™ $E - A - R$ LINK Insert Eartips) were used for air-conducted rarefaction stimulation (500 Hz short tone bursts were presented at 5 per second with a rise/fall and plateau time of 2-2-2 ms), and one control trial was run at 80 dB nHL, and a minimum of two trials were run at 100 dB nHL. The latencies and amplitudes for p13 and n23 peaks were recorded for each ear.

### Ocular VEMP (oVEMP)

Participants were seated upright and asked to keep a 30° upward gaze. The active electrode was placed 0.5 cm below the eye, parallel to the lateral half of the lower eyelid, and the medial corner of the active electrode was placed below the eye—at the midline of the eye. A reference electrode was placed on the upper part of the forehead, and the ground electrode below the reference electrode. A bone conductor (B-81 modified with a double headband to deliver more energy to the bone; Interacoustics®, Middlefart, Denmark. 500 Hz short tone bursts were presented at 5 bursts per second with a rise/fall and plateau time of 2-2-2 ms) was placed on the mastoid process and used for alternating polarity stimulation in one

control trial at 50 dB nHL and a minimum of two trials at 70 dB nHL. The latencies and amplitudes for n10 and p15 peaks were recorded for each ear.

## Statistical analysis

Statistical analyses were conducted using Stata I/C version 14.2. (StataCorp, USA). The level of significance was set at $p < 0.05$. Continuous covariates were compared by the Wilcoxon rank sum test, and normally distributed data were compared by a $t$-test. Categorical variables were compared by the Chi-square test. Interpersonal variance was tested by Bland–Altman plots and normal distribution of the data was tested by reviewing graphical distributions and each continuous variable was tested for normality of distribution by the Shapiro–Wilk test. The sum of sway was calculated for all eight positions and as the sum of the four neutral (no pillow eyes open ST, no pillow eyes closed ST, pillow eyes open ST, pillow eyes closed ST) and four head tilt/turn (head right, head left, head back, head forward). This study includes data on secondary outcomes of a subpopulation from a previously published study, and therefore a sample size calculation was not conducted prior to the study.

# RESULTS

A total of 107 individuals were included in the present study and consisted of three groups: type 2 diabetes individuals with DPN ($n = 43$), type 2 diabetes individuals without DPN ($n = 32$), and healthy control individuals ($n = 32$). Within the past year, individuals with type 2 diabetes reported a higher number of falls compared to healthy controls ($p = 0.04$), however there was no significant difference in the number of reported falls when comparing individuals with and without DPN ($p = 0.71$) (Table 1). Data on age, body weight, height, BMI and waist circumference were normally distributed ($p > 0.05$), however data on diabetes duration, HbA1c levels and postural stability did not follow a normal distribution ($p < 0.01$ for all).

When comparing individuals with type 2 diabetes to healthy controls, no difference was found in age. However, individuals with type 2 diabetes had a higher body weight, an increased BMI, waist circumference, and had increased postural instability when compared to healthy controls (Table 1). An increased diabetes duration, increased HbA1c levels, increased use of insulin, and increased postural instability were found in individuals with DPN compared to individuals with diabetes without DPN (Table 1).

Amplitudes and latencies on the right and left side were not normally distributed ($p < 0.01$, for all). Comparing all individuals with diabetes to healthy controls and individuals with diabetes with and without DPN, no difference was found for the other cVEMP and oVEMP measurements (Table 2).

Comparing all individuals with diabetes to healthy controls, there was a greater number of no-responses in oVEMP ($p = 0.04$), irrespective of DPN. No difference was found in the number of no responses in cVEMP (Table 2).

## Fallers *vs.* non-fallers

In Table 3, VEMP parameters, for left and right ears from individuals with type 2 diabetes and with falls ($n = 27$) *versus* no falls ($n = 48$), are presented. Fallers had shorter oVEMP

**Table 1  Clinical and biochemical characteristics.**

| | Control Individuals | Individuals with type 2 diabetes | | | |
| | n = 32 | without DPN n = 32 | with DPN n = 43 | p-value[a] | p-value[b] |
| --- | --- | --- | --- | --- | --- |
| Age, years | 62 (8) | 63 (9) | 64 (7) | 0.41 | 0.67 |
| Female gender (n,(%)) | 14 (44) | 17 (53) | 13 (30) | | |
| Height (cm) | 175 (7) | 169 (8) | 174 (10) | 0.14 | 0.02 |
| Weight (kg) | 86 (18) | 90 (18) | 103 (18) | 0.01 | 0.01 |
| BMI (kg/m2) | 28 (5) | 31 (6) | 34 (6) | 0.01 | 0.06 |
| *Waist circumference* | | | | | |
| Females (cm) | 97 (24) | 105 (14) | 118 (15) | 0.03 | 0.02 |
| Males (cm) | 103 (9) | 113 (11) | 120 (12) | 0.01 | 0.09 |
| *Diabetes profile* | | | | | |
| Diabetes duration (years) | NA | 7 (6; 10) | 10 (6; 18) | | 0.02 |
| HbA1c, (mmol/mol) | 37 (34; 39) | 48 (45; 55) | 56 (48; 68) | 0.01 | 0.02 |
| Insulin (Yes) (n, (%)) | NA | 3 (9) | 22 (51) | | 0.01 |
| Oral anti-diabetes agents (n,(%)) | NA | 27 (84) | 38 (88) | | 0.65 |
| Fallers (n, (%)) | 5 (16) | 12 (38) | 15 (35) | 0.04 | 0.71 |
| *Instability index* | | | | | |
| Average ST in neutral positions | 23 (17; 28) | 28 (22; 33) | 41 (29; 64) | 0.01 | 0.01 |
| Average ST in tilt/turn positions | 28 (22; 38) | 34 (25; 42) | 60 (38; 94) | 0.01 | 0.01 |
| Average ST in all positions | 26 (19; 33) | 31 (24; 39) | 52 (33; 77) | 0.01 | 0.01 |

**Notes.**

NA, Not Applicable; categorical data are frequencies (%); continuous data are medians (p25, p75). Continuous covariates were compared by the Wilcoxon rank sum test, and normally distributed data were compared by a *t*-test. Categorical variables were compared by the Chi square test.

[a] *p*-value comparing individuals with diabetes and healthy controls.

[b] *p*-value comparing individuals with diabetes without DPN and individuals with DPN.

DPN, diabetic polyneuropathy; BMI, body mass index; ST, stability index.

(n10) latencies and cVEMP (p13) compared to non-fallers, significant for right ears and with a similar tendency for left ears, however not significant. In contrast, no significant difference was found in cVEMP (p13-n23) and oVEMP (n10-p15) amplitudes, nor cVEMP (n23) latencies. No difference was found in the number of no responses in cVEMP and oVEMP when comparing fallers and non-fallers.

## DISCUSSION

This cross-sectional study examined the association between vestibular dysfunction, falls, and postural instability in individuals with type 2 diabetes, with and without DPN, compared to healthy controls. Individuals with diabetes reported more falls within the previous year, irrespective of DPN. Individuals with diabetes had increased postural instability, which was even more pronounced in individuals with DPN. In individuals with type 2 diabetes, fallers had shorter oVEMP (n10) and cVEMP (p13) latencies on right ears compared to non-fallers, irrespective of DPN. Similar tendencies were seen for left

**Table 2  Vestibular-evoked myogenic potential parameters for right and left ears in each group.**

| | Control Individuals | Individuals with type 2 diabetes | | p-value[a] | p-value[b] |
|---|---|---|---|---|---|
| | $n = 32$ | without DPN $n = 32$ | with DPN $n = 43$ | | |
| **Right ear** | | | | | |
| cVEMP p13 (ms) | 16 (14; 17) | 15 (14; 16) | 16 (14; 18) | 0.73 | 0.19 |
| cVEMP n23 (ms) | 24 (23; 26) | 24 (21; 26) | 25 (22; 26) | 0.73 | 0.49 |
| oVEMP n10 (ms) | 16 (14; 29) | 16 (14; 29) | 21 (16; 31) | 0.18 | 0.20 |
| oVEMP p15 (ms) | 11 (10; 23) | 11 (10; 25) | 15 (11; 25) | 0.11 | 0.08 |
| cVEMP (p13- n23) (μV) | 61 (30; 87) | 65 (33; 87) | 35 (24; 73) | 0.39 | 0.09 |
| cVEMP (p13- n23) SCALED | 1 (0; 1) | 1 (0; 1) | 1 (0; 1) | 0.63 | 0.11 |
| oVEMP (n10-p15) (μV) | 10 (5; 20) | 11 (7; 25) | 8 (5; 14) | 0.98 | 0.10 |
| **Left ear** | | | | | |
| cVEMP p13 (ms) | 15 (15; 16) | 15 (14; 16) | 16 (13; 17) | 0.58 | 0.18 |
| cVEMP n23 (ms) | 24 (23; 26) | 24 (23; 25) | 25 (23; 26) | 0.80 | 0.26 |
| oVEMP n10 (ms) | 17 (15; 28) | 16 (14; 35) | 17 (0; 28) | 0.82 | 0.50 |
| oVEMP p15 (ms) | 11 (10; 23) | 11 (10; 29) | 12 (0; 24) | 0.78 | 0.63 |
| cVEMP (p13- n23)(μV) | 66 (37; 94) | 55 (41; 83) | 47 (21; 70) | 0.09 | 0.14 |
| cVEMP (p13- n23) SCALED | 1 (0; 1) | 1 (1; 1) | 0 (0; 1) | 0.36 | 0.10 |
| oVEMP (n10-p15) (μV) | 8 (6; 19) | 13 (6; 21) | 7 (2; 17) | 0.91 | 0.28 |
| No response in total oVEMP (n, (%)) | 9 (28) | 15 (47) | 25 (58) | 0.04 | 0.33 |
| No response in total cVEMP (n, (%)) | 4 (13) | 4 (13) | 7 (16) | 0.78 | 0.65 |

Notes.

[a]p- value comparing individuals with diabetes and healthy controls.

[b]p-value comparing individuals with diabetes without DPN and individuals with DPN. Continuous data are medians (p25, p75); categorical data are frequencies (%).

DPN, diabetic polyneuropathy; cVEMP, cervical vestibular-evoked myogenic potential; oVEMP, ocular vestibular-evoked myogenic potential; ms, millisecond; μV, microvolts.

SCALED: the VEMP amplitude is scaled/normalizedin proportion to the tonic EMG activity. (Averaged VEMP response amplitude (μV) divided by root mean square of pre-stimulationEMG activity (μV)).

ears, however not significant. Individuals with type 2 diabetes had a greater number of no-responses in oVEMP compared to healthy controls, irrespective of DPN.

To our knowledge, this is the first study examining vestibular dysfunction using cVEMP and oVEMP in relation to falls and postural instability in individuals with type 2 diabetes, with and without diabetic polyneuropathy, compared to healthy controls.

In previous studies of falls and postural instability, individuals with diabetes and DPN had more vestibular dysfunction combined with an increased risk of falls (*Agrawal et al., 2009*; *Agrawal et al., 2010*). Additionally, individuals with diabetes with and without DPN had more postural instability compared to healthy controls. In contrast to our findings, fallers with diabetes had a greater incidence of peripheral neuropathy (*Khan et al., 2021*; *Allet et al., 2014*; *Emam et al., 2009*; *Vaz et al, 2013*; *MacGilchrist et al., 2010*). Balance is a complex skill requiring the cooperation of somatosensory, vestibular, and visual systems together with muscular and cognitive systems (*Hewston & Deshpande, 2015*). In individuals with diabetes, the causes of falls are most likely multifactorial as diabetes may affect one

**Table 3  Vestibular-evoked myogenic potential parameters for left and right ears in fallers and non-fallers with diabetes.**

| | Individuals with type 2 diabetes | | |
| --- | --- | --- | --- |
| | **No Falls** **n = 48** | **≥1** **Fall n = 27** | ***p*-value** |
| *Right ears* | | | |
| cVEMP p13 (ms) | 16 (15; 17) | 15 (8; 16) | 0.04 |
| cVEMP n23 (ms) | 25 (22; 26) | 23 (13; 26) | 0.16 |
| oVEMP n10 (ms) | 25 (16; 35) | 16 (15; 19) | 0.03 |
| oVEMP p15 (ms) | 22 (11; 30) | 11 (10; 12) | 0.14 |
| cVEMP (p13- n23) (μV) | 59 (29; 80) | 37 (13; 80) | 0.32 |
| cVEMP (p13- n23) SCALED | 1 (0; 1) | 0 (0; 1) | 0.39 |
| oVEMP (n10-p15) (μV) | 8 (7; 18) | 9 (7; 14) | 0.97 |
| **No responses** | | | |
| oVEMP (n, (%)) | 23 (48) | 19 (68) | 0.09 |
| cVEMP (n, (%)) | 11(15) | 7 (21) | 0.40 |
| *Left ears* | | | |
| cVEMP p13 (ms) | 16 (14; 17) | 16 (7; 17) | 0.92 |
| cVEMP n23 (ms) | 24 (23; 26) | 24 (11; 26) | 0.73 |
| oVEMP n10 (ms) | 16 (14; 37) | 15 (7; 19) | 0.20 |
| oVEMP p15 (ms) | 12 (10; 31) | 11 (5; 13) | 0.31 |
| cVEMP (p13-n23) (μV) | 51 (31; 66) | 55 (11; 91) | 0.60 |
| cVEMP (p13-n23) SCALED | 1 (0; 1) | 1 (0; 1) | 0.95 |
| oVEMP (n10-p15) (μV) | 11 (6; 20) | 5 (2; 18) | 0.30 |
| **No-responses** | | | |
| oVEMP (*n*, (%)) | 27 (56) | 20 (71) | 0.19 |
| cVEMP (*n*, (%)) | 10 (13) | 7 (21) | 0.30 |

**Notes.**
Continuous data are medians (p25, p75); categorical data are frequencies (%).
cVEMP, cervical vestibular-evoked myogenic potential; oVEMP, ocular vestibular-evoked myogenic potential; ms, millisecond; μV, microvolts.
SCALED: The VEMP amplitude isscaled/normalizedin proportion to the tonicEMGactivity. (Averaged VEMP response amplitude (μV) divided by root mean square of pre-stimulationEMG activity (μV)).

or more of these systems (*Hewston & Deshpande, 2015*). In our study, individuals with diabetes had a greater number of no-responses on the oVEMP. No-responses can indicate impaired nerve function which is seen in more advanced stages of nerve dysfunction (*Taylor et al., 2020*). However, the number of no-responses was not greater in the DPN group.

Several studies have examined vestibular function using VEMP in individuals with type 2 diabetes (*Ward et al., 2015*; *Konukseven et al., 2015*; *Kalkan et al., 2018*), in individuals with type 1 diabetes (*Kamali et al., 2013*), and in individuals with non-insulin-dependent diabetes mellitus (*Bektas et al., 2008*). Some of these studies only examined the vestibular function using cVEMP and not oVEMP, which is inadequate as cVEMP is believed to reflect the ipsilateral sacculus and inferior vestibular nerve function, whereas oVEMP reflect the contralateral utriculus and superior vestibular nerve function (*Rosengren &*

*Kingma, 2013*). In our study, we examined the vestibular function using both cVEMP and oVEMP providing more details on the vestibular function.

Studies on VEMP responses in individuals with diabetes show conflicting results. This might be attributed to the smaller sample sizes, heterogeneity in the used cVEMP and oVEMP tests, and a lack of homogeneity of clinical and biochemical characteristics, including age, diabetes duration, and HbA1c levels (*El Bakkali et al., 2021*). In contrast to previous studies (*Ward et al., 2015*; *Kalkan et al., 2018*), we did not find any differences in cVEMP amplitudes and oVEMP amplitudes. Other previous studies (*Konukseven et al., 2015*; *Bektas et al., 2008*; *Kamali et al., 2013*) found no differences in cVEMP (p13-n23) and oVEMP (n10-p15) amplitudes. Previous studies (*Konukseven et al., 2015*; *Kamali et al., 2013*) found prolonged cVEMP (p13-n23) latencies in individuals with diabetes compared to healthy controls. In contrast to these findings, but in line with other studies (*Ward et al., 2015*; *Bektas et al., 2008*; *Kalkan et al., 2018*), we found no differences in cVEMP (p13 and n23) latencies. Many of our study participants have newly diagnosed diabetes with only a mild degree of DPN. This can possibly explain, why we found no differences in any of the VEMP parameters when comparing all three groups and when comparing individuals with diabetes with and without DPN.

In diabetic animals, various structural and functional changes in the vestibular system have been found, including overproduction of extracellular matrix and increased lipid droplets in the otolith organs, degeneration of type 1 hair cells, thinning of the myelin covering the vestibulocochlear nerve and smaller diameter of the axonal fibers (*Myers & Ross, 1987*; *Myers, Tormey & Akl, 1999*). Human studies have shown abnormalities of the vestibulo-ocular and optokinetic reflex, and deficits in gaze-holding in individuals with type 2 diabetes compared to individuals without type 2 diabetes (*Nicholson et al., 2002*). These structural and functional changes may compromise vestibular information leading to inadequate motor responses and thereby, ultimately a fall. Vestibular dysfunction was more prevalent in individuals with diabetes (*Agrawal et al., 2009*), and vestibular dysfunction was shown independently to increase the odds of falling more than two times, even after adjusting for diabetic polyneuropathy (DPN) (*Agrawal et al., 2010*). In our study, fallers with diabetes exhibited poorer vestibular function compared to non-fallers with diabetes.

## Limitations and strengths

There are several limitations to our study: (1) Due to the cross-sectional design, we cannot determine if the association between diabetes and vestibular function is causal; (2) numbers of falls within the past year was based solely on the recollection of participants. This might have introduced recall bias leading to incorrect numbers of falls. However, we chose one year to rule out seasonal influence on fall incidences (*Magota et al., 2017*). (3) Only individuals fending for themselves and living relatively close to Aarhus University Hospital were included, which probably has left out individuals with more severe diabetes thereby introducing selection bias.

The main strengths of our study are: (1) The same physician examined all individuals regarding VEMP-testing, measuring of sway, and clinical DPN assessment, which secured consistency in the examinations, (2) DPN diagnosis was confirmed by both nerve

conduction studies and clinical examination and (3) reliable, and validated methods were applied in the examination of the vestibular function and postural stability (*Christensen et al., 2018*; *Nguyen, Welgampola & Carey, 2010*; *Isaradisaikul et al., 2008*). Furthermore, a significant strength of our study is the use of VEMP-testing for measuring vestibular function being a direct assessment of the vestibular function. Other studies (*Agrawal et al., 2009*; *Agrawal et al., 2010*) have used the modified Romberg Test of Standing Balance. This testing tool is a poor screening tool for vestibular dysfunction compared to VEMP-testing (*Jacobson et al., 2011*).

Contrary to our study, some previous studies (*Allet et al., 2014*; *Emam et al., 2009*; *Vaz et al, 2013*; *Hewston & Deshpande, 2018*; *Brown et al., 2015*), assessing falling and postural instability in individuals with diabetes and DPN, did not include a healthy control group or did not compare results between individuals with diabetes with and without DPN, which impairs the evaluation of the impact of both diabetes and DPN per se. Some studies did not clearly define a fall or did not exclude other causes of falls.

Future studies should include larger sample sizes with a prospective study design. Furthermore, future studies should consider using VEMP-testing for the evaluation of vestibular function and its relation to fall incidents in individuals with longer diabetes duration and more severe disease.

In summary, falls and postural instability were more frequent in individuals with type 2 diabetes compared to healthy controls. No-responses for the oVEMP latencies were more frequent in individuals with type 2 diabetes compared to healthy controls, demonstrating impaired vestibular end nerve function, irrespective of DPN.

### Funding
The research reported in this publication is part of the International Diabetic Neuropathy Consortium (IDNC) research programme, which is supported by a Novo Nordisk Foundation Challenge Programme grant (Grant number NNF14OC0011633) and Aarhus University. Aarhus University receives funding for other studies from companies in the form of research grants to (and administered by) Aarhus University. None of these studies has any relation to the present study. The funders had no role in study design, data collection and analysis, decision to publish, or preparation of the manuscript.

### Grant Disclosures
The following grant information was disclosed by the authors:
International Diabetic Neuropathy Consortium (IDNC) Research Programme.
Novo Nordisk Foundation Challenge Programme:  NNF14OC0011633.

### Competing Interests
The authors declare there are no competing interests.

## Author Contributions

- Ina Ejsing Hunnerup Jørgensen analyzed the data, prepared figures and/or tables, authored or reviewed drafts of the article, and approved the final draft.
- Louise Devantier conceived and designed the experiments, performed the experiments, prepared figures and/or tables, authored or reviewed drafts of the article, and approved the final draft.
- Hatice Tankisi conceived and designed the experiments, performed the experiments, authored or reviewed drafts of the article, and approved the final draft.
- Henning Andersen conceived and designed the experiments, analyzed the data, authored or reviewed drafts of the article, and approved the final draft.
- Karolina Snopek Khan conceived and designed the experiments, performed the experiments, analyzed the data, prepared figures and/or tables, authored or reviewed drafts of the article, and approved the final draft.

## Ethics

The following information was supplied relating to ethical approvals (i.e., approving body and any reference numbers):

The study protocol was registered with the Danish Data Protection Agency (approval no.:1-16-02-563-16) and approved by the Central Denmark Region Committees on Health Research Ethics (approval no.: 1-10-72-282-16).

## Data Availability

The raw measurements are available in the Supplementary Files.

## Supplemental Information

Supplemental information for this article can be found online at http://dx.doi.org/10.7717/peerj.16382#supplemental-information.

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
