# Peer review of "The impact of vestibular dysfunction on falls and postural instability in individuals with type 2 diabetes with and without diabetic polyneuropathy"

_PeerJ, doi:10.7717/peerj.16382_

## Round 0.1 · original submission · Major Revisions

The authors should address the concerns raised by the reviewers and improve the information, including statistics, to make their experimental design clearer. Conclusions should be considered in the light of the results obtained.

·

Basic reporting

The manuscript is presented with a good contextual framework, however, there are grammatical errors that deserve a review by a fluent English speaker. Moreover, the results provided do not clearly confirm the hypotheses put forward by the authors.

Experimental design

The manuscript is aligned with the aims and scope of the journal and complies with the ethical requirements for research on human subjects. However, some of the procedures decreed in the methodology need to be deepened and the statistical analyses performed need to be reviewed in depth.

Validity of the findings

The manuscript should be improved in terms of information related to recording procedures and statistical analysis to ensure replicability. Given these comments, the conclusions of the paper are not entirely clear.

·

Basic reporting

Most important issues:
• Description of cut-off values or when abnormality is seen in oVEMP and cVEMP testing and balance testing.
• Description of inclusion criteria (see below)
• Grammar / spelling

Strengths:
• Good academic English, very easy to read
• DPN assessment
• Association between falling, vestibular function and DPN
• Clear data tables

Experimental design

Large inclusion of participants compared to previous research for oVEMP/cVEMP testing in diabetes.

Validity of the findings

No comment

Additional comments

No comment.

---

## Round 0.2 · Major Revisions

Please provide a complete point-by-point revision to the issues raised by both reviewers.

**Language Note:** The review process has identified that the English language must be improved. PeerJ can provide language editing services - please contact us at copyediting@peerj.com for pricing (be sure to provide your manuscript number and title). Alternatively, you should make your own arrangements to improve the language quality and provide details in your response letter. – PeerJ Staff

·

Basic reporting

The manuscript is presented with a good contextual framework, however, there are grammatical errors that still deserve a review by a fluent English speaker. Moreover, the clarity of the results seems to me debatable given some statistical procedures that require attention.

Experimental design

The manuscript is aligned with the aims and scope of the journal and complies with the ethical requirements for research on human subjects. However, some of the procedures decreed in the methodology need to be deepened and the statistical analyses performed need to be reviewed in depth

Validity of the findings

The manuscript should be improved in terms of information related to recording procedures and statistical analysis to ensure replicability.

·

Basic reporting

Please rewrite lines 39-41, it reads as an unfinished sentence.

Please reread line 81 and 83, check spelling.

Lines 131-136 should be described under results section.

Please provide reference for lines 250 - 252 (This line was added by you after my peer-review last time. I had attached a review that concluded this, thus you may add it as a reference to support your statement): https://doi.org/10.1016/j.deman.2021.100035

Experimental design

Good, no comment

Validity of the findings

Good, no comment

Additional comments

Very minor revision needed for some spelling, a sentence not placed in the right paragraph and add a reference to a statement in your discussion.

---

## Round 0.3 · Minor Revisions

The authors are invited to address the few remaining issues raised by the reviewers.

·

Basic reporting

no comment

Experimental design

no comment

Validity of the findings

I would first like to thank the authors for their efforts in taking responsibility for the comments made. For my part, only one doubt persists regarding the statistical analysis section. Although the authors mention that they used the Wilcoxon test to determine the differences, the previous step of applying a statistical test to determine the distribution of the data through a hypothesis test and not only through a visual inspection of the distribution is still missing. I believe that this is a relevant step to ensure the replicability of the information and should therefore be made explicit.

·

Basic reporting

No new remarks regarding content of the manuscript and data. Only regarding spelling:

- Check lines 238 and 254.

Experimental design

None

Validity of the findings

None

Additional comments

None

---

## Round 0.4 · accepted · Accept

The authors have satisfactorily addressed the remaining issues raised by the reviewers, so that in the present form the manuscript can be accepted for publication.

·

Basic reporting

no comment

Experimental design

no comment

Validity of the findings

no comment

·

Basic reporting

Professional, clear English is used. No further comments.

Experimental design

No further comments.

Validity of the findings

No further comments.

Additional comments

No further comments.